# Cycling reduces the entropy of neuronal activity in the human adult cortex

Iara Beatriz Silva Ferré[1], Gilberto Corso[2], Gustavo Zampier dos Santos Lima[3], Sergio Roberto Lopes[4], Mario André Leocadio-Miguel[5]*, Lucas G. S. França[6,7], Thiago de Lima Prado[4], John Fontenele Araújo[1]

1 Programa de Pós-Graduação em Psicobiologia, Universidade Federal do Rio Grande do Norte, Natal, RN, Brazil, 2 Departamento de Biofísica e Farmacologia, Universidade Federal do Rio Grande do Norte, Natal, RN, Brazil, 3 Escola de Ciência e Tecnologia, Universidade Federal do Rio Grande do Norte, Natal, RN, Brazil, 4 Departamento de Física, Universidade Federal do Paraná, Curitiba, PR, Brazil, 5 Department of Psychology, Northumbria University, Newcastle upon Tyne, United Kingdom, 6 Department of Computer and Information Sciences, Faculty of Engineering and Environment, Northumbria University, Newcastle upon Tyne, United Kingdom, 7 Department of Forensic and Neurodevelopmental Science, Institute of Psychiatry, Psychology & Neuroscience, King's College London, London, United Kingdom

* mario.miguel@northumbria.ac.uk

**Data Availability Statement:** Data acquired in the context of this study, the code used to generated the results, figures and statistics in this article are available in the Open Science Framework repository at https://doi.org/10.17605/OSF.IO/

## Abstract

Brain Complexity (BC) have successfully been applied to study the brain electroencephalographic signal (EEG) in health and disease. In this study, we employed recurrence entropy to quantify BC associated with the neurophysiology of movement by comparing BC in both resting state and cycling movement. We measured EEG in 24 healthy adults and placed the electrodes on occipital, parietal, temporal and frontal sites on both the right and left sides of the brain. We computed the recurrence entropy from EEG measurements during cycling and resting states. Entropy is higher in the resting state than in the cycling state for all brain regions analysed. This reduction in complexity is a result of the repetitive movements that occur during cycling. These movements lead to continuous sensorial feedback, resulting in reduced entropy and sensorimotor processing.

## Introduction

Walking involves important cognitive processes governing neural networks incorporating sensory information, motor planning, execution and sensorial feedback [1]. Whilst an essential component of human life, this complexity is the reason our walking capacity is greatly sensitive to ageing [2] and can be severely compromised in neurological disorders [3]. Compared to walking, cycling comprises a cyclical and repetitive movement, consisting of the complete rotation of the pedal axis around the central axis of the bicycle. While the motor planning and execution of cycling might have similarities to the cortical control of walking, it is a less cognitively demanding movement. For instance, the need for trunk balance is significantly reduced when compared to walking [4]. This discrepancy motivated the investigation of the EEG signal between the resting state, cycling and walking. Storzer et al. [5] contrasted the EEG dynamics involved in cycling and walking in healthy volunteers. Both cycling and walking resulted in an

CW87P and at https://github.com/AlgosL/cycling-entropy.git. The script for entropy estimation is available at https://github.com/AlgosL/maxEntropy.

**Funding:** The author(s) received no specific funding for this work.

**Competing interests:** The authors have declared that no competing interests exist.

increased recruitment of cortical activity compared to resting state. Nevertheless, cycling exhibited reduced recruitment for the execution of the movement and, therefore, corresponding to simpler dynamics, compared to walking.

Changes in cortical neuron recruitment have been previously analysed with brain complexity (BC) metrics derived from complex system science. Those studies provided insights into processes in healthy individuals and a number of conditions [6]. One such metric is *entropy*, which, when applied to gauge EEG signals, will provide a marker of the information encoded in those, i.e. changes in neuronal recruitment will modify electrical signals recorded by each electrode and that can be quantified with entropy. For example, a reduction in complexity is evident during slow wave sleep or when the eyes are closed during wakefulness [7, 8]. Entropy has also been proposed as a tool to distinguish the EEG of Parkinson's disorder patients from controls [9] and to identify the transitions from walking to freezing in such patients [10].

This study investigated BC, gauged by entropy, and its association with the neurophysiological context during cycling in healthy volunteers. We contrasted the scalp electroencephalogram (EEG) of individuals while cycling and at rest. Here we hypothesise that cycling itself does reduce BC when compared to rest.

## Materials and methods

### Experimental setup

**Sample.** The sample consisted of 24 adult volunteers, 13 women (54%; mean age = 21.30; SE = 0.49) and 11 men (46%; mean age = 21.63; SE = 0.87). We performed an assessment of the subjects through an interview about the clinical conditions and measurements of blood pressure and heart rate before the experiment. Subjects who reported any cardio-respiratory or neurological disease or who had blood pressure values above 90 mmHg and 140 mmHg, for diastolic and systolic blood pressure respectively, were excluded from the sample. We also excluded individuals who showed artefacts in the recorded EEG. From the initial 24 individuals, 6 individuals were excluded. The excluded individuals showed erratic jumps in the electric signal, which are typical artefacts that result from poor skin-electrode contact.

**Legal and ethical aspects.** The present study was analysed by the Research Ethics Committee (CEP) of Universidade Federal do Rio Grande do Norte, and approved under CAAE 02979318.0.0000.5537. In order to participate in our project, the volunteers expressed their consent to participate in the research by signing the informed consent form (written consent) according to the Helsinki Declaration. The study recruited participants from 1 March to 14 June 2019.

**Procedure and electroencephalographic evaluation.** We used electroencephalography to acquire cortical electrical activity data, which records the electrical brain activity with electrodes fixed to the individual's scalp. Participants started with a 2-minute baseline rest period sitting on the bicycle. Subsequently, they cycled continuously for 2 minutes. We acquired those with both open and closed eyes to assess the effect of alpha rhythms on BC [11]. For signal acquisition, Ag-AgCl electrodes were positioned on the scalp according to the 10-20 system. An electroencephalographic assembly was composed of eight electrodes in a bimodal assembly, the following pairs being F3-Fz, F4-Fz, C3-Cz, C4-Cz, P3-Pz, P4-Pz, O1-A2 and O2-A1. To facilitate the understanding, we named the electrodes as follows: F, C, P and O, in addition to the left and right sides. For the placement of the electrodes, the subjects only had to stop using hair creams on the day of the experiment to facilitate fixation of the EEG cap on the scalp and data collection. An abrasive paste for skin asepsis was used in the fixation sites of each electrode, and all electrodes were affixed with a conductive paste at each pre-established site of the scalp.

Data were collected at a sampling rate of 1000 Hz using a PowerLab 8/30 system (AdInstruments, Australia). ECG data were recorded on a PowerLab 26T system. Both systems were integrated, and the data wwere recorded using Labchart 7 Pro Software (AdInstruments, Australia). Data were captured with a band-pass filter from 1 to 100 Hz. For analysis, a band-stop filter of 59-61 Hz was applied to remove noise from the electrical network and, during cycling activity, a band-pass filter of 3 to 35 Hz was applied.

**The bicycle model.** There are three main types of stationary bikes: horizontal, upright and spinning. In this experiment we used a horizontal bicycle, which is generally used as a form of aerobic exercise for cardiac rehabilitation, weight loss, and as a form of stress testing. The horizontal bicycle reduces possible sensory interference and the risk of impacts for individuals who participate in the activity, such as the risk of falls. Moreover, the individuals cycling in the horizontal bike are more stable and as a consequence the skin electrodes have a better adhesion. In fact, the electrophysiology artefacts related to a weak skin electrode adhesion have a great impact on the quality of the measurements [12]. The bicycle model used in this project is the MAX-H by Dream Fitness (Brazil).

## Data

From each individual, a total of 32 electrophysiologic recordings were obtained. These recordings correspond to eight electrode signals measured in four distinct behavioral states. The eight electrodes correspond to the places O, P, F and C for right and left sides. Moreover, each participant was registered at rest and cycling, with both open and closed eyes.

## Entropy

Out of several complexity indices used to measure complexity of time series in previous research, e.g. entropy, Lyapunov exponent, or fractal dimension [13–16], entropy is a long-standing tool to explore complex phenomena [17] and can be defined according to the relation in Eq 1.

$$S = \sum_i p_i \, log \, p_i \tag{1}$$

for $p_i$ a probability associated with the time series. In the original work of Shannon [18], the probability is associated with information carried by the time series. In neuroscience literature, the probability is usually related to the amplitude of the spectral representation of the signal [19]. We computed entropy using the recurrence method [20, 21]. This novel technique has been successfully employed to capture signal complexity by evaluating the periodicity of a time series and recurrence analysis has been widely employed in dynamical systems research and time series analyses [22], as it requires smaller data amounts to return reliable results [23, 24]. In our case, we split each electrophysiologic recording into 100 pieces to compute the recurrence indices.

## Statistics

We evaluated the statistical significance of our results with general linear models (GLM), because they are flexible and can handle a variety of data types and distributions, which is typical of EEG.

First, we assessed the effect of open and closed eyes on cortical entropy according to GLM1: *MedianEntropy* ∼ *Eyes*(*Open*/*Closed*) + (1|*Subject*). This first analysis was performed as a validation step and compared with previous results reported elsewhere.

We then fitted a second general linear mixed effects model (GLMM) due to repeated measures considering the cycling effect according to GLMM2: *MedianEntropy ~ State(Cycling/Rest)\*Eyes(Open/Closed) + (1|Subject)*.

The analyses were performed with R version 4.3.2 [25] and auxiliary packages [26, 27]. In both cases, the statistical significance of the variables of interest was evaluated with two-sided 10,000 repetitions permutation tests as they require fewer assumptions regarding the samples and their distributions [28]. We established $\alpha = 0.05$ as the significance level.

## Results

### Reduced cortical entropy with eyes closed

To validate the recurrence entropy methodology, we compared the signal complexity in a region of the brain where we expect a strong signal response. The entropy of the electrode signal from the right side of the occipital region is shown in Fig 1. In this situation, all individuals

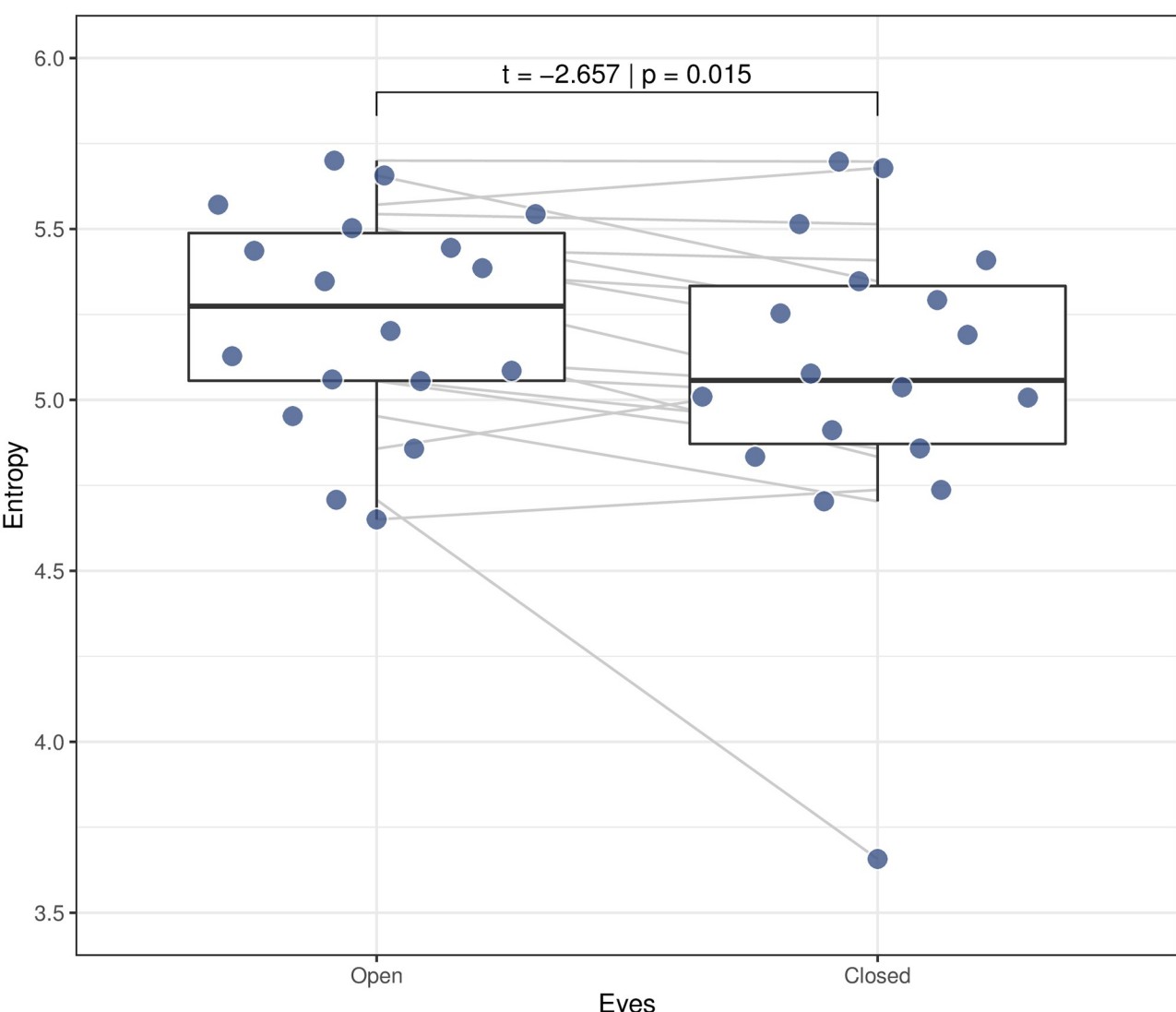

**Fig 1. Variation of median entropy with closed eyes.** A reduced cortical entropy is associated with closed eyes; the variation is statistically significant as evaluated with the GLM1: *MedianEntropy ~ Eyes(Open/Closed) + (1|Subject)* and a two-sided 10,000 repetitions permutation test.

are at rest, and we just compared eyes open and closed. The open eye shows larger entropy compared to closed eyes $t(16) = -2.657$; $p = 0.015$.

## Cycling is associated with reduced cortical entropy

We then analysed the effect of cycling in the EEG signal. The cycling state shows lower entropy when compared with the resting state for left and right central ($t = -3.824$; $p = 0.001$), left and right frontal ($t = -3.617$; $p = 0.001$ and $t = -2.164$; $p = 0.042$, respectively), left and right occipital ($t = -3.337$; $p = 0.002$ and $t = -2.901$; $p = 0.006$, respectively), and left and right parietal ($t = -4.754$; $p < 0.001$ and $t = -2.497$; $p = 0.018$, respectively) regions, shown in Fig 2A.

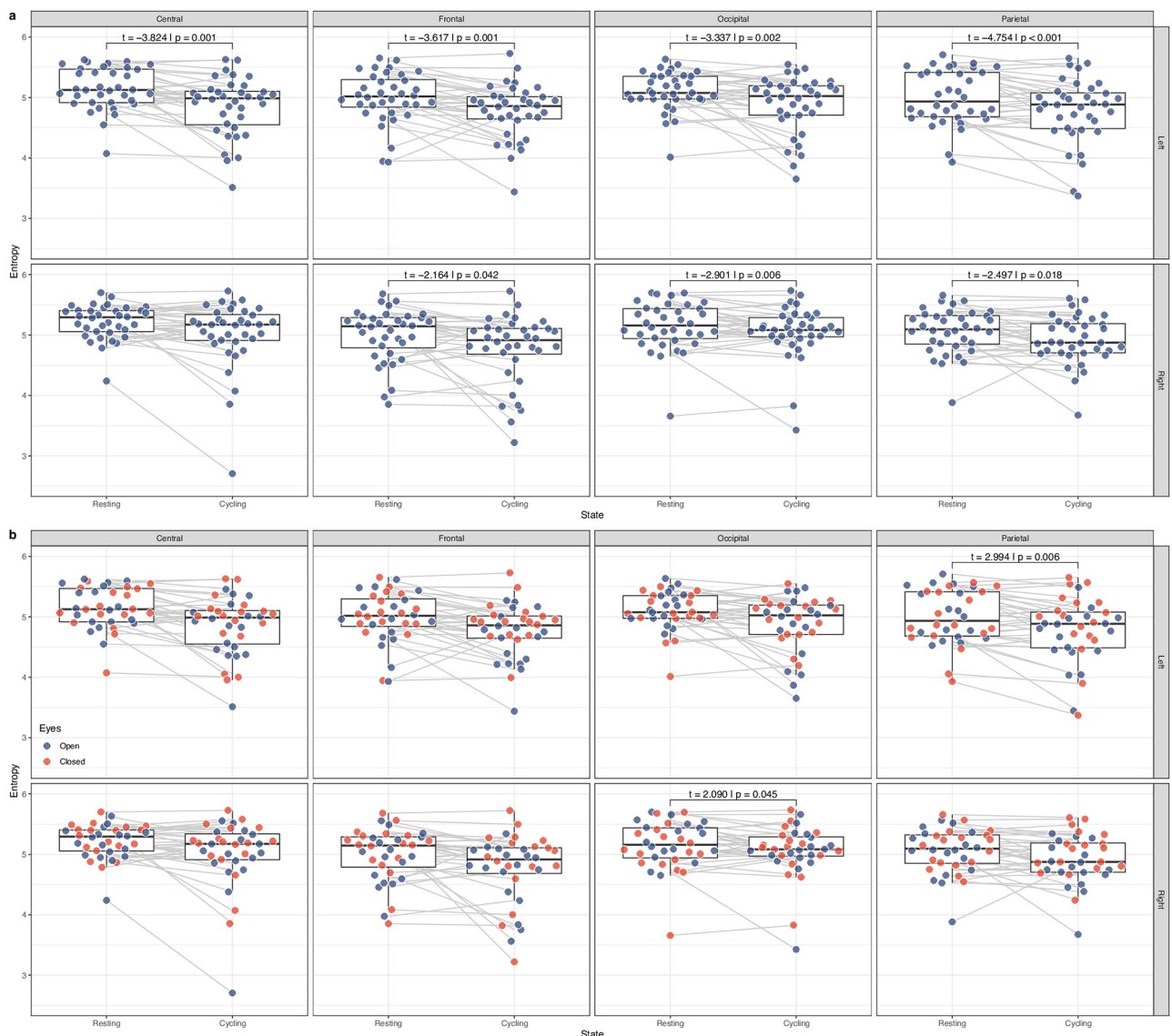

**Fig 2. Variation of median entropy with cycling.** a) A reduced cortical entropy is associated with cycling. b) Measures for closed and open eyes session. A reduced cortical entropy is also associated with interaction between cycling and having one's eyes closed. Reported values are statistically significant as evaluated with the GLMM2: $MedianEntropy \sim State(Cycling/Rest)*Eyes(Open/Closed) + (1|Subject)$ and a two-sided 10,000 repetitions permutation test.

### Reduced cortical entropy whilst cycling with closed eyes

To evaluate the effect of cycling with closed eyes, we have also analysed the interaction term for our model for closed eyes and cycling. The cycling state with eyes closed shows lower entropy for right occipital ($t = 2.090$; $p = 0.045$) and left parietal ($t = −2.994$; $p = 0.006$) regions, shown in Fig 2B. A topographic plot with the entropy profile for the four different scenarios is shown in Fig 3.

## Discussion

We applied recurrence entropy [20, 21] as a measure of complexity to assess changes in cortical electrophysiological activity associated with distinct brain functional states. The methodology for estimating signal complexity via entropy of recurrence was initially validated using the electrophysiological signal of the occipital region and comparing the signal entropy between open-eyes and closed-eyes. The result showed a reduction in entropy in the closed-eyes condition. It is known that the open-eyes condition allows retinal light detection which leads to neural activity in the occipital region responsible for processing visual information, that is represented by desynchronised neuronal activity. When we close our eyes and thus prevent light stimuli, an intrinsic neural activity of the thalamus-cortex circuit dominates the signal with a predominant frequency between 8 and 12 Hz, known as the alpha rhythm [11, 29, 30]. Our results suggest that the complexity measurement we propose is capable of distinguishing both: a synchronised pattern with lower entropy (closed eyes) and a desynchronised pattern with higher entropy (open eyes).

The aim of our study was to evaluate electrophysiological changes between rest and cycling conditions. Using the same recurrence entropy, we showed that the neural complexity is lower during the cycling behavior when compared to the resting state. This reduction in entropy was observed with greater predominance in the anterior areas of the brain, especially in the frontal area. We suggest that the observed entropy reduction is associated with an increase in cortical synchronization due to the increase of sensory feedback originating from the lower limbs as a result of flexion and extension repetitive movements that alternately occur during cycling. Our results are in line with those from Storzer et al. [5], who demonstrated a different pattern in the dynamics of neuronal oscillation in the cortex associated with walking and cycling behaviors compared to the resting state. Cycling behavior reduces the neuronal activity at high frequencies, in the beta band range (20 to 30 Hz) during the movement, followed by a rebound of this beta activity when exercise terminates [5]. In this way, decreased beta band power has been linked to an active neural state in the sensorimotor cortex that is associated with an increase in cortical excitability [31]. Furthermore, it has been shown that the beta power remains suppressed during continuous movements [32], but it is maintained in isometric and sustained movements [33, 34]. As the cycling is continuous, this movement may cause a stronger decrease in beta power, and it would explain the entropy reduction found in our study.

Spectral analysis is widely used by neuroscientists and indeed the identification of brain waves is built upon frequency bands [35]. Nevertheless, this technique has some important drawbacks, for example the dependence on noise that pollutes the frequency spectrum, often making it difficult to identify the dominant frequencies [36]. In addition, the EEG signal is non-linear and non-stationary with a high degree of complexity [37]. Therefore, the use of the Fourier transform is not entirely appropriate [7]. Furthermore, the spectral analysis is dependent on some arbitrary factors of choice: the size and shape of the window to analyse the signal and the type of base used. We point out that the basis of sines and cosines associated with the Fourier transform is the most common basis, but the Wavelet transforms open the way to many alternative bases [38]. In recent years, with the availability of mathematical tools based

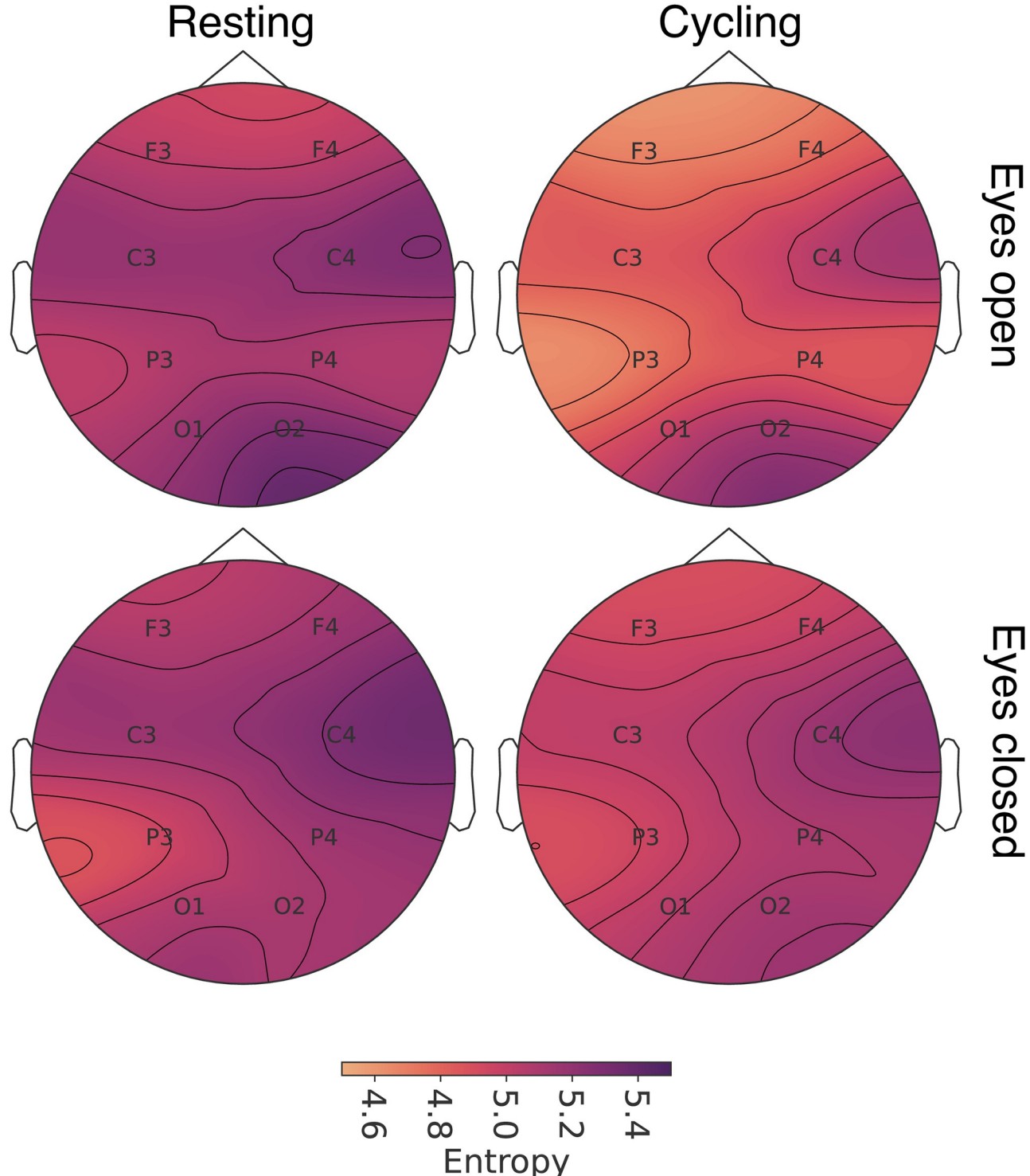

**Fig 3. Topographic plots for median entropy on F3-Fz, F4-Fz, C3-Cz, C4-Cz, P3-Pz, P4-Pz, O1-A2 and O2-A1 electrodes.** The profiles show a reduction in entropy figures for eyes closed and cycling states—with the lowest values registered for cycling with eyes closed.

on complexity theories [15, 16], we have observed the use of entropy-based approaches as a strategy for nonlinear EEG analysis to provide independent and complementary measurements to conventional EEG spectral analysis, and with this, it has been possible to characterise these entropy measures with discrete changes in behavioral states [6, 39]. In this context, the recurrence entropy is a tool that directly uses the time series, without the need for preprocessing involving spectral analysis, to estimate the complexity of an electrophysiological signal.

In addition to exploring how cycling interferes with the complexity of brain electrophysiology in healthy individuals, our motivation for performing this study is based on recent research which demonstrated that cycling instantly decreases Parkinson's disease (PD) motor signals while PD patients suffering from difficulties in walking and freezing of gait ride a bicycle [40]. The degeneration in the substantia nigra, the hallmark of PD, can lead to the inability to take a step associated with short steps that normally occur at the beginning of gait or when turning during walking, greatly impairing the mobility of patients, and resulting in the increased risk of falls and drastically reducing their quality of life [41]. As additional impacts, dynamic cycling increases sensory input to the motor control of movement in Parkinson's disease patients, which may be related to improvements in motor speed and quality [42]. In future work, we intend to use this technique to assess brain complexity in different cycling conditions, such as cycling on horizontal and vertical stationary bicycles or cycling in a virtual reality environment. Furthermore, we aim to study the brain complexity of healthy, proficient and non-proficient cyclists and PD patients, using a stationary bicycle as a tool.

## Conclusion

Our results clearly indicate that the entropy of neuronal activity in the human adult cortex decreases during cycling movements. This reduction in complexity is due to the repetitive movements that occur during cycling, which may cause continuous sensory feedback, resulting in less entropy and sensorimotor processing.

## Acknowledgments

We would like to thank Mr Matthew Osbourn for supporting with the proofreading of this article.

## Author Contributions

**Conceptualization:** Iara Beatriz Silva Ferré, Gilberto Corso, John Fontenele Araújo.

**Formal analysis:** Gilberto Corso, Gustavo Zampier dos Santos Lima, Sergio Roberto Lopes, Lucas G. S. França, Thiago de Lima Prado.

**Investigation:** Iara Beatriz Silva Ferré, John Fontenele Araújo.

**Methodology:** Gilberto Corso, Gustavo Zampier dos Santos Lima, John Fontenele Araújo.

**Project administration:** John Fontenele Araújo.

**Software:** Gilberto Corso, Sergio Roberto Lopes, Thiago de Lima Prado.

**Supervision:** John Fontenele Araújo.

**Writing – original draft:** Mario André Leocadio-Miguel, John Fontenele Araújo.

**Writing – review & editing:** Gilberto Corso, Gustavo Zampier dos Santos Lima, Mario André Leocadio-Miguel, Lucas G. S. França, John Fontenele Araújo.

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
