## [Decision Letter · Decision Letter 0]

5 Mar 2024

PONE-D-24-02271Cycling reduces the entropy of neuronal activity in the human adult cortexPLOS ONE

Dear Dr. Leocadio-Miguel,

Thank you for submitting your manuscript to PLOS ONE. After careful consideration, we feel that it has merit but does not fully meet PLOS ONE’s publication criteria as it currently stands. Therefore, we invite you to submit a revised version of the manuscript that addresses the points raised during the review process.

**ACADEMIC EDITOR:**

The paper is interesting, however, major revisions are required. Please revise thoroughly upon reviewers' comments and submit the revised version. 

We look forward to receiving your revised manuscript.

Kind regards,

Noman Naseer, PhD

Academic Editor

PLOS ONE

A clean copy of the edited manuscript (uploaded as the new *manuscript* file).

“The authors thank the Brazilian Council for Scientific and Technological Development (CNPq) for its continued support. Grants Nos. 309440/2022-0, 308441/2021-4, 305189/2022-0, grant 307907/2019-8.”

4. Please upload a copy of Supporting Information Figure 1 which you refer to in your text on page 4.

Additional Editor Comments:

The paper is interesting, however, major revisions are required. Please revise thoroughly upon reviewers' comments and submit the revised version.

Reviewers' comments:

Reviewer's Responses to Questions

**Comments to the Author**

1. Is the manuscript technically sound, and do the data support the conclusions?

Reviewer #1: Yes

Reviewer #2: Partly

2. Has the statistical analysis been performed appropriately and rigorously? 

Reviewer #1: I Don't Know

Reviewer #2: Yes

3. Have the authors made all data underlying the findings in their manuscript fully available?

Reviewer #1: Yes

Reviewer #2: Yes

4. Is the manuscript presented in an intelligible fashion and written in standard English?

Reviewer #1: Yes

Reviewer #2: No

5. Review Comments to the Author

Reviewer #1: The topic is interesting and should be beneficial for further research. The conclusion is not in cohesion with the other part of the manuscript and should be rewritten in the form of answering the main research question.

Reviewer #2: Review for “Cycling reduces the entropy of neuronal activity in the human adult cortex”

I would like to appreciate the unique idea behind this paper. There are certain amendments which I would want the authors to consider while revisiting this manuscript:

1. The abstract attempts to covers all the aspects of the topic, which is good. However, I do think that the first sentence which introduces EEG and the second sentence which instantly starts talking about the present study and recurrence entropy, without really making the connection.

2. When talking about Regions of Interest, the authors need to mention brain when saying ‘right and left sides’.

3. There is no connection created between entropy and cycling and the logic of connecting cycling with EEG or the very idea of doing this research. Why do we want to know about entropy? How is it beneficial?

4. The introduction is not impressive, as the authors directly begin with something that does not directly correlate academically with the title. The starting statement says “Richard Caton once said that the brain speaks for itself”, which is not impressive for a research paper.

5. Starting with the history of EEG is not a good idea, in the introduction. I would recommend to start with your main topic and why is it needed.

6. The comparison of walking with cycling holds interesting dynamics, however, it seems slightly inconclusive at the end on page 8.

7. From cycling, there is a sudden jump to gait without really defining and linking it with cycling.

8. The introduction section misses numerous links between ideas. How entropy impacts human beings, why would brain behave differently, why do we need to assess it, where can it be beneficial, why are we exploring it in cycling, why is there a need to do this?

9. Details of Sample is needed. What was the criteria for selecting these 13 women and 11 men?

10. The authors used the horizontal bicycle for the experiment. There is a need for citation here to justify the use of this bicycle in particular.

11. There is a need for a visual in the Methodology section which could exemplify the experimental paradigm and trials.

12. The logic, citation and justification for using eyes close and open is missing in the methodology section.

13. In the discussion section, the authors suddenly move from cycling to walking and its neuronal dynamics, which seems haphazard and requires to be connected well.

14. In the conclusion section, the topic has been linked with Parkinson’s disease at least 4 to 5 times. However, the sample taken for this study did not include this disease and its concerning manifestations. This makes the conclusion confusing for the reader to navigate. The authors need to revise this section with particular attention.

15. The statistical analysis requires more justification and details. Presently, it is very surficial and does not cover many reasons of how and why it was performed.

16. Many of the references are not according to the format. They need to be rechecked.

6. PLOS authors have the option to publish the peer review history of their article (what does this mean?). If published, this will include your full peer review and any attached files.

Reviewer #1: No

Reviewer #2: No

---

## [Author Response · Author response to Decision Letter 0]

14 Jun 2024

Rebuttal Letter - Cycling reduces the entropy of neuronal activity in the human adult cortex – Ferre et al, PLOSOne

The authors would like to thank the reviewers for their fair reviews and detailed list of constructive suggestions. We greatly appreciate the opportunity to submit a major revision of our paper for your renewed consideration. Below, we respond to reviewers’ comments and point to the corresponding changes we made in the paper. The line numbers below refer to the line numbers in the clean revised manuscript. We have included an extra file containing all the track changes.

Reviewer #1: 

Comment: The topic is interesting and should be beneficial for further research. The conclusion is not in cohesion with the other part of the manuscript and should be rewritten in the form of answering the main research question.

Response: We would like to thank the reviewer for the comments and feedback, which allowed us to improve the quality of our manuscript. We emphasised reorganising the manuscript to answer the main research question clearly. 

Action: We reworked the Conclusion section to address the issue highlighted by the reviewer. Accordingly, our new conclusion section is as follows (lines 213-217):

“Our results clearly indicate that the entropy of neuronal activity in the human adult cortex decreases during cycling movements. This reduction in complexity is due to the repetitive movements that occur during cycling, which may cause continuous sensory feedback, resulting in less entropy and sensorimotor processing.”

Reviewer #2: 

Review for “Cycling reduces the entropy of neuronal activity in the human adult cortex”

Comment: I would like to appreciate the unique idea behind this paper. There are certain amendments which I would want the authors to consider while revisiting this manuscript:

Response: We would like to thank the reviewer for the feedback. We tried our best to address every single concern raised by the reviewer. 

Comment: 1. The abstract attempts to covers all the aspects of the topic, which is good. However, I do think that the first sentence which introduces EEG and the second sentence which instantly starts talking about the present study and recurrence entropy, without really making the connection.

Response: We would like to thank the reviewer for the feedback. 

Action: We reworked the abstract section to address the issue highlighted by the reviewer. Accordingly, our new abstract section is as follows:

“Brain Complexity (BC) have successfully been applied to study the brain electroencephalographic signal (EEG) in health and disease. In this study, we employed recurrence entropy to quantify BC associated with the neurophysiology of movement by comparing BC in both resting state and cycling movement. We measured EEG in 24 healthy adults and placed the electrodes on occipital, parietal, temporal and frontal sites on both the right and left sides of the brain. We computed the recurrence entropy from EEG measurements during cycling and resting states. Entropy is higher in the resting state than in the cycling state for all brain regions analysed. This reduction in complexity is a result of the repetitive movements that occur during cycling. These movements lead to continuous sensorial feedback, resulting in reduced entropy and sensorimotor processing.”

Comment: 2. When talking about Regions of Interest, the authors need to mention brain when saying ‘right and left sides’.

Response: Thank you for highlighting this detail. 

Action: We have adjusted the sentence in the abstract section to clarify it. 

Comment: 3. There is no connection created between entropy and cycling and the logic of connecting cycling with EEG or the very idea of doing this research. Why do we want to know about entropy? How is it beneficial?

Response: Thank you for bringing this detail to our attention.

Action: By reorganising the abstract section, we stated the benefits of using brain complexity measures to process EEG data in health and disease. We also emphasised the connection between reduced entropy and sensorimotor processing.

Comment: 4. The introduction is not impressive, as the authors directly begin with something that does not directly correlate academically with the title. The starting statement says “Richard Caton once said that the brain speaks for itself”, which is not impressive for a research paper.

Response: Thank you for the comment. Our idea was to provide the reader with a historical background, highlighting the gap between the development of the first EEG devices and the more recent and currently underdeveloped assessment of brain activity during movement. 

Action: However, to make our introduction section more focused, we decided to exclude the entire first paragraph. 

Comment: 5. Starting with the history of EEG is not a good idea, in the introduction. I would recommend to start with your main topic and why is it needed.

Response: Thank you for the comment. 

Action: As described in the response to the previous comment, we reorganised the introduction section. Therefore, the new first paragraph is as follows (lines 2-16):

“Walking involves important cognitive processes governing neural networks incorporating sensory information, motor planning, execution and sensorial feedback [1]. Whilst an essential component of human life, this complexity is the reason our walking capacity is greatly sensitive to ageing [2] and can be severely compromised in neurological disorders [3]. Compared to walking, cycling comprises a cyclical and repetitive movement, consisting of the complete rotation of the pedal axis around the central axis of the bicycle. While the motor planning and execution of cycling might have similarities to the cortical control of walking, it is a less cognitively demanding movement. For instance, the need for trunk balance is significantly reduced when compared to walking [4]. This discrepancy motivated the investigation of the EEG signal between the resting state, cycling and walking. Storzer et al. [5] contrasted the EEG dynamics involved in cycling and walking in healthy volunteers. Both cycling and walking resulted in an increased recruitment of cortical activity compared to resting state. Nevertheless, cycling exhibited reduced recruitment for the execution of the movement and, therefore, corresponding to simpler dynamics, compared to walking.”

Comment: 6. The comparison of walking with cycling holds interesting dynamics, however, it seems slightly inconclusive at the end on page 8.

Response: We appreciate the comment and addressed the issue in the introduction section. 

Action: We paid extra care to address this comparison in the last two sentences of the first paragraph (lines 13-16). 

Comment: 7. From cycling, there is a sudden jump to gait without really defining and linking it with cycling.

Response: Thank you for the relevant comment. 

Action: The new structure of the introduction section is straightforward. It privileges the differences in the dynamics of cycling and walking in the first part and presents entropy to assess brain complexity in the second half. This reorganisation was meant to be more coherent and to deal with the sudden jumps between concepts. 

Comment: 8. The introduction section misses numerous links between ideas. How entropy impacts human beings, why would brain behave differently, why do we need to assess it, where can it be beneficial, why are we exploring it in cycling, why is there a need to do this?

Response: We would like to thank the reviewer for the thoughtful comment, which was essential as a guide for the new structure of the introduction section. 

Action: We reorganised the second paragraph to provide readers with the links and rationale behind adopting entropy to study EEG dynamics. Therefore, the new second paragraph is as follows (lines 17-26):

“Changes in cortical neuron recruitment have been previously analysed with brain complexity (BC) metrics derived from complex system science. Those studies provided insights into processes in healthy individuals and a number of conditions [6]. One such metric is entropy, which, when applied to gauge EEG signals, will provide a marker of the information encoded in those, i.e. changes in neuronal recruitment will modify electrical signals recorded by each electrode and that can be quantified with entropy. For example, a reduction in complexity is evident during slow wave sleep or when the eyes are closed during wakefulness [7,8]. Entropy has also been proposed as a tool to distinguish the EEG of Parkinson’s disorder patients from controls [9] and to identify the transitions from walking to freezing in such patients [10].”

Comment: 9. Details of Sample is needed. What was the criteria for selecting these 13 women and 11 men?

Response: Thank you for the comment. We recruited our volunteers amongst healthy undergraduate students via leaflets and word-of-mouth within the academic community. These individuals volunteered to participate in the study and received no incentives. We aimed at a sample size compatible with that of Storzer et al., who recruited 15 healthy volunteers and maintained 14 after exclusion criteria. Our final sample count was reached after applying the exclusion criteria described in the methodology.

Action: We have clarified the exclusion criteria in the methodology section to explain better how we reached the final sample. The new first paragraph of the materials and methods section is as follows (lines 34-43):

“The sample consisted of 24 adult volunteers, 13 women (54%; mean age = 21.30; SE = 0.49) and 11 men (46%; mean age = 21.63; SE = 0.87). We performed an assessment of the subjects through an interview about the clinical conditions and measurements of blood pressure and heart rate before the experiment. Subjects who reported any cardio-respiratory or neurological disease or who had blood pressure values above 90 mmHg and 140 mmHg, for diastolic and systolic blood pressure respectively, were excluded from the sample. We also excluded individuals who showed artefacts in the recorded EEG. From the initial 24 individuals, 6 individuals were excluded. The excluded individuals showed erratic jumps in the electric signal, which are typical artefacts that result from poor skin-electrode contact.”

Comment: 10. The authors used the horizontal bicycle for the experiment. There is a need for citation here to justify the use of this bicycle in particular.

Response: The rationale behind the adoption of this bicycle is based on the fact that the recumbent or horizontal posture has the potential to minimise the impact of mechanical artefacts on the EEG signal, as described by Bailey, S. P., Hall, E. E., Folger, S. E., & Miller, P. C. (2008), reference number 12. 

Action: We included the following reference in the respective manuscript section (line 82): Bailey, S. P., Hall, E. E., Folger, S. E., & Miller, P. C. (2008). Changes in EEG During Graded Exercise on a Recumbent Cycle Ergometer. Journal of Sports Science & Medicine, 7(4), 505–511.

Comment: 11. There is a need for a visual in the Methodology section which could exemplify the experimental paradigm and trials.

Response: Thank you for the suggestion. We have adjusted the respective section in the manuscript to clarify the experimental procedure. 

Action: The new version of the respective paragraph follows (lines 52-56): “We used electroencephalography to acquire cortical electrical activity data, which records the electrical brain activity with electrodes fixed to the individual’s scalp. Participants started with a 2-minute baseline rest period sitting on the bicycle. Subsequently, they cycled continuously for 2 minutes. We acquired those with both open and closed eyes to assess the effect of alpha rhythms on BC [11].”

Comment: 12. The logic, citation and justification for using eyes close and open is missing in the methodology section.

Response: We thank the reviewer. We acknowledge the missing citation and have provided it in the revised version of the manuscript. 

Action: We have included further explanation in the methodology to make it clearer for the reader (lines 55 - 56). “We acquired those with both open and closed eyes to assess the effect of alpha rhythms on BC”

This is also further explained in the discussion section (lines 148 – 156): “The methodology for estimating signal complexity via entropy of recurrence was initially validated using the electrophysiological signal of the occipital region and comparing the signal entropy between open-eyes and closed-eyes. The result showed a reduction in entropy in the closed-eyes condition. We know that the open-eyes condition allows retinal light detection which leads to neural activity in the occipital region responsible for processing visual information, that is represented by desynchronised neuronal activity. When we close our eyes and thus prevent light stimuli, an intrinsic neural activity of the thalamus-cortex circuit dominates the signal with a predominant frequency between 8 and 12 Hz, known as the alpha rhythm [11, 29, 30].”

Comment: 13. In the discussion section, the authors suddenly move from cycling to walking and its neuronal dynamics, which seems haphazard and requires to be connected well.

Response: We want to thank the reviewer for the insightful suggestion. Based on this feedback, we reorganised the discussion section, reducing the sudden changes and coherence issues and narrowing the focus towards a more results-oriented narrative. 

Action: We have excluded the sentences regarding comparing walking and cycling.

Comment: 14. In the conclusion section, the topic has been linked with Parkinson’s disease at least 4 to 5 times. However, the sample taken for this study did not include this disease and its concerning manifestations. This makes the conclusion confusing for the reader to navigate. The authors need to revise this section with particular attention.

Response: Once again, thank you for the perfect comment. In fact, the original conclusion was not appropriate. 

Action: We have changed the conclusions section entirely. The reorganised version is straightforward and focused on our findings (lines 213-217). 

Comment: 15. The statistical analysis requires more justification and details. Presently, it is very surficial and does not cover many reasons of how and why it was performed.

Response: Thank you. We acknowledge that further details were necessary. Therefore, we changed the manuscript accordingly. 

Action: Specifically, we included details on the suitability of general linear models to handle data extracted from EEG recordings. We also improved the description of the second GLM model and included the significance level of the statistical procedures. Finally, we described and included a citation referring to the rationale for the permutation test (lines 104-118). 

Comment: 16. Many of the references are not according to the format. They need to be rechecked.

Response: We would like to thank the reviewer for this comment. 

Action: As a result, we have double-checked the reference list to guarantee the references comply with the format. 

Again, thank you for the opportunity to submit a revised version of our manuscript. Your comments and suggestions were fundamental to improving the quality of our study's report. 

Kindest regards,

Mario Leocadio-Miguel

Corresponding author

---

## [Decision Letter · Decision Letter 1]

18 Jul 2024

Cycling reduces the entropy of neuronal activity in the human adult cortex

PONE-D-24-02271R1

Dear Dr. Leocadio-Miguel,

We’re pleased to inform you that your manuscript has been judged scientifically suitable for publication and will be formally accepted for publication once it meets all outstanding technical requirements.

Kind regards,

Noman Naseer, PhD

Academic Editor

PLOS ONE

Additional Editor Comments (optional):

The reviewers' concerns have been adequetly addressed.

Reviewers' comments:

Reviewer's Responses to Questions

**Comments to the Author**

1. If the authors have adequately addressed your comments raised in a previous round of review and you feel that this manuscript is now acceptable for publication, you may indicate that here to bypass the “Comments to the Author” section, enter your conflict of interest statement in the “Confidential to Editor” section, and submit your "Accept" recommendation.

Reviewer #2: All comments have been addressed

2. Is the manuscript technically sound, and do the data support the conclusions?

Reviewer #2: Yes

3. Has the statistical analysis been performed appropriately and rigorously? 

Reviewer #2: Yes

4. Have the authors made all data underlying the findings in their manuscript fully available?

Reviewer #2: Yes

5. Is the manuscript presented in an intelligible fashion and written in standard English?

Reviewer #2: Yes

6. Review Comments to the Author

Reviewer #2: I have checked the amendments made in the draft the by the authors. These revisions satisfy my comments and I think the manuscript is fine now.

7. PLOS authors have the option to publish the peer review history of their article (what does this mean?). If published, this will include your full peer review and any attached files.

Reviewer #2: No

---

## [Editor Report · Acceptance letter]

6 Sep 2024

PONE-D-24-02271R1 

PLOS ONE

Dear Dr. Leocadio-Miguel, 

I'm pleased to inform you that your manuscript has been deemed suitable for publication in PLOS ONE. Congratulations! Your manuscript is now being handed over to our production team.

Kind regards, 

on behalf of

Dr. Noman Naseer 

Academic Editor

PLOS ONE